# βH-spectrin is required for ratcheting apical pulsatile constrictions during tissue invagination

Daniel Krueger*, Cristina Pallares Cartes, Thijs Makaske & Stefano De Renzis**

## Abstract

Actomyosin-mediated apical constriction drives a wide range of morphogenetic processes. Activation of myosin-II initiates pulsatile cycles of apical constrictions followed by either relaxation or stabilization (ratcheting) of the apical surface. While relaxation leads to dissipation of contractile forces, ratcheting is critical for the generation of tissue-level tension and changes in tissue shape. How ratcheting is controlled at the molecular level is unknown. Here, we show that the actin crosslinker βH-spectrin is upregulated at the apical surface of invaginating mesodermal cells during *Drosophila* gastrulation. βH-spectrin forms a network of filaments which co-localize with medio-apical actomyosin fibers, in a process that depends on the mesoderm-transcription factor Twist and activation of Rho signaling. βH-spectrin knockdown results in non-ratcheted apical constrictions and inhibition of mesoderm invagination, recapitulating *twist* mutant embryos. βH-spectrin is thus a key regulator of apical ratcheting during tissue invagination, suggesting that actin cross-linking plays a critical role in this process.

**Keywords** actomyosin; apical constriction; optogenetics; tissue invagination; tissue morphogenesis

**Subject Categories** Cell Adhesion, Polarity & Cytoskeleton; Development

## Introduction

Cell shape changes driven by contraction of cortical actomyosin filaments are of fundamental importance during animal development, underlying key morphogenetic processes such as cytokinesis (Pollard, 2010; Sedzinski *et al*, 2011), cell migration (Blanchoin *et al*, 2014), and localized remodeling of tissue shape (Bertet *et al*, 2004; Dawes-Hoang *et al*, 2005; Butler *et al*, 2009; Fernandez-Gonzalez *et al*, 2009; Heisenberg & Bellaiche, 2013; Izquierdo *et al*, 2018; Krueger *et al*, 2018). Fast *in vivo* imaging and *in vitro* studies demonstrate that contraction of cortical actomyosin networks is controlled by pulsatile flows of myosin-II molecules, which move centripetally as actin filaments contract and cell surface shrinks (Martin *et al*, 2009; Solon *et al*, 2009; Reymann *et al*, 2012;

Banerjee *et al*, 2017). Myosin-II pulses are followed by either relaxation or stabilization of the cell surface, before a new contractile cycle starts. While it remains unclear whether pulsatile behavior plays an active role during morphogenesis, stabilization of the cell surface, usually referred to as ratcheting, has been suggested to be critical for generation of supracellular forces and large-scale tissue remodeling (Martin *et al*, 2009; Solon *et al*, 2009; Clement *et al*, 2017). According to this view, ratcheting might prevent dissipation of forces, which would otherwise occur during relaxation of the cell surface, thus facilitating force transmission through adherens junctions and building of tissue-level tension. Ratcheting might also serve as a mechanism to protect tissues from tearing apart as cells constrict (Ducuing & Vincent, 2016). As yet however, how ratcheting is controlled at the molecular level remains poorly understood (Munjal *et al*, 2015; Sumi *et al*, 2018; Miao *et al*, 2019). Here, we have identified the actin crosslinker βH-spectrin as a key regulator of apical ratcheting during *Drosophila* ventral furrow invagination.

This morphogenetic process is driven by pulsatile apical constrictions and cell shape changes of a group of ~ 1,000 cells arranged in a rectangular pattern on the ventral surface of the embryo (Martin *et al*, 2009; Guglielmi *et al*, 2015). Apical constriction in ventral cells requires expression of the transcription factors Twist and Snail, which also confer mesodermal fate to the invaginating cells (Leptin, 1991). Twist and Snail control expression of several downstream targets (Rembold *et al*, 2014) converging on Rho signaling and myosin-II activation (Dawes-Hoang *et al*, 2005; Kolsch *et al*, 2007). However, while in *snail* mutant embryos cells do not apically constrict and change shape, in *twist* mutants cells undergo uncoordinated cycles of apical constriction and relaxation without maintaining the constricted state, suggesting that molecular mechanisms downstream of Twist control ratcheting of the apical surface (Martin *et al*, 2009). These contractile alterations result in impaired tissue invagination and are phenocopied by treating embryos with low doses of actin depolymerizing drugs. In both, *twist* mutants and cytochalasin D-treated embryos, the network of medio-apical actomyosin filaments that forms at the onset of ventral furrow formation (see cartoon in Fig 1A–C) either does not assemble or when present it loses its attachment to the junctions causing cells to expand abnormally when neighboring cells constrict (Mason *et al*, 2013).

The results presented in this study show that the actin crosslinker βH-spectrin is upregulated at the apical surface of mesodermal cells

---

Developmental Biology Unit, European Molecular Biology Laboratory (EMBL), Heidelberg, Germany
*Corresponding author. Tel: +49 6221 387 8109; Fax: +49 6221 387 8166; E-mail: daniel.krueger@embl.de
**Corresponding author. Tel: +49 6221 387 8109; Fax: +49 6221 387 8166; E-mail: derenzis@embl.de

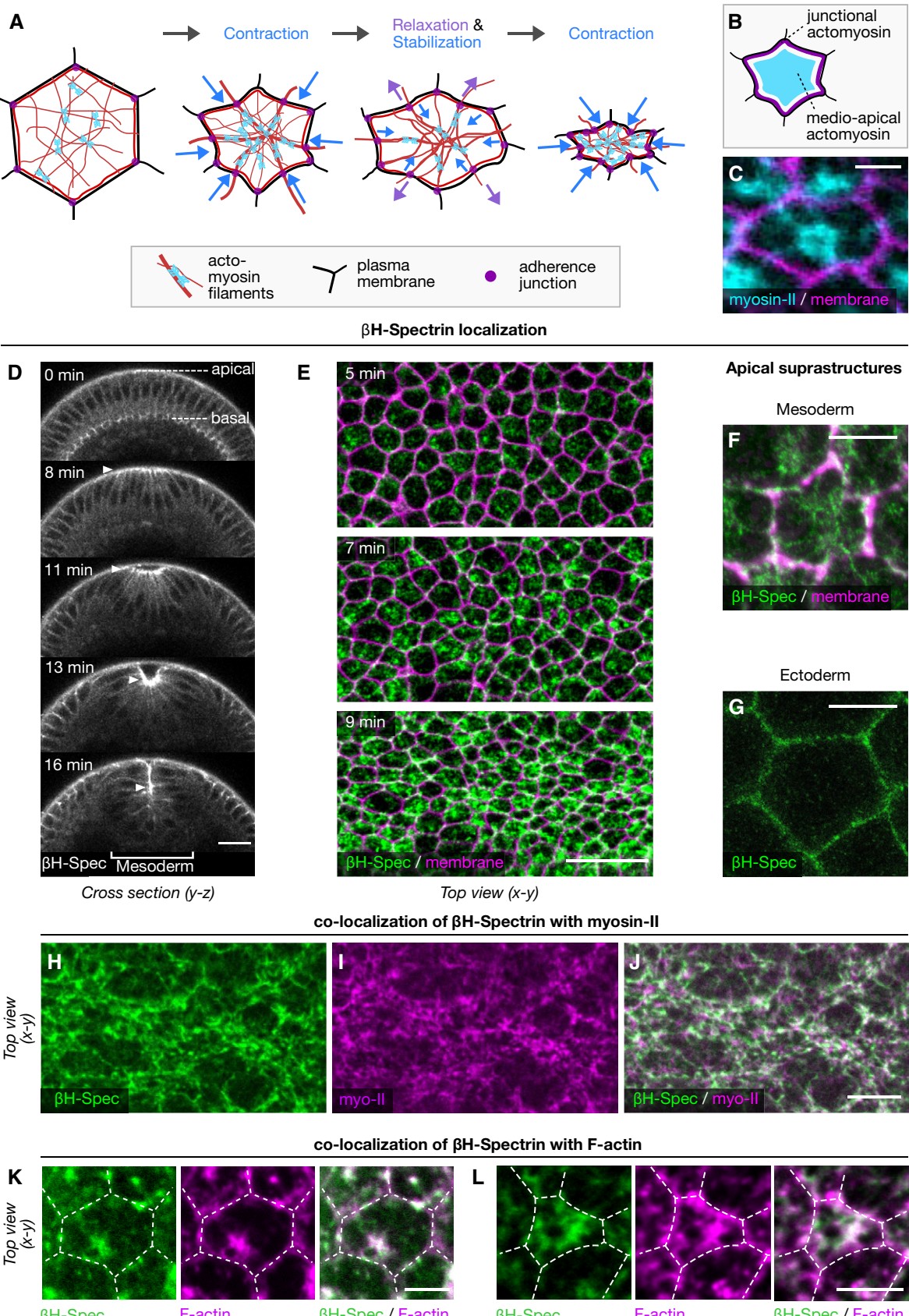

**Figure 1.**

**Figure 1. βH-spectrin co-localizes with myosin-II to medio-apical fibers in mesodermal cells during ventral furrow formation.**

A, B   Schematic illustration of a constricting cell undergoing a cycle of ratcheted pulsatile constriction. Upregulation of myosin-II (blue) causes coalescence of actin fibers (red) in a radially polarized manner. Contraction of medio-apical actomyosin fibers, which are anchored to the plasma membrane (black) through adherens junctions (purple dots) and are mechanically coupled to a junctional actomyosin belt (red), causes constriction of the cell surface. Disassembly of medial actomyosin is accompanied by surface relaxation, which is contained by a ratcheting mechanism that stabilizes the constricted state. Repeated cycles of actomyosin-mediated ratcheted contraction result in an incremental shrinkage of the cell surface. (B) Schematic illustrating medio-apical (cyan) and junctional (purple) actomyosin fibers.

C   Confocal image of a constricting cell during ventral furrow formation showing an overlay between the myosin-II probe Sqh::GFP and the plasma membrane marker GAP43::mCherry. Scale bar, 5 μm.

D   Movie (still frames) of a gastrulating *Drosophila* embryo expressing endogenously tagged mVenus::βH-spectrin imaged in a cross section using two-photon microscopy. βH-spectrin is enriched at the apical surface of ventral mesodermal cells during tissue invagination (arrowheads). Scale bar, 50 μm.

E   Z-Projections of confocal image stacks showing the apical cell surface of a *Drosophila* embryo expressing endogenously tagged mVenus::βH-spectrin (green) and the plasma membrane marker GAP43::mCherry at 5 min (top), 7 min (middle) and 9 min (bottom) after initial medio-apical accumulation of βH-spectrin. Scale bar, 20 μm

F, G   Immunostaining of βH-spectrin (green) visualized by STED nanoscopy revealed medio-apical βH-spectrin supracellular fibers in mesodermal cells (F). In ectodermal cells, βH-spectrin localizes to apical cell junctions (G). Note, in (F) the junctional βH-spectrin signal is shown in magenta as a proxy for cell membranes. Scale bars, 2.5 μm.

H–J   Confocal images of the ventral surface of a *Drosophila* embryo expressing endogenously tagged mVenus::βH-spectrin (H), the myosin-II marker Sqh::mCherry (I) and a merge of the two (J) with βH-spectrin in green and Sqh::mCherry in magenta. Scale bars: 20 μm.

K, L   Co-staining of phalloidin (F-actin reporter; magenta) and βH-spectrin (green) of the apical surface of a mesodermal cell at the onset of ventral furrow formation (K) and at a later time point (L) visualized by confocal microscopy. White dashed lines indicate the cell boundaries segmented based on the phalloidin staining of sub-apical confocal sections. Scale bars: 5 μm.

during ventral furrow invagination in a process that requires the zygotic expression of *twist* and Rho signaling activation. βH-spectrin localizes to medio-apical actomyosin fibers, and its activity is required for ratcheting apical constrictions as demonstrated by nano-body-mediated protein knockdown. Similar to the *twist* mutant phenotype, reducing βH-spectrin protein levels does not inhibit apical constrictions. Rather it causes cells to pulse without stabilizing the apical surface resulting in defects in tissue invagination and integrity. Together these results support a model in which apical ratcheting during tissue invagination is controlled by βH-spectrin-dependent actin cross-linking and surface organization.

## Results and Discussion

We have recently characterized a mechanism based on actin cross-linking that regulates the contraction of a basally localized acto-myosin network during cellularization, a morphogenetic process that immediately precedes ventral furrow invagination (Krueger *et al*, 2019). In particular, we found that bottleneck, a protein expressed for only a short time during cellularization, bundles actin filaments and counteracts actomyosin contractility allowing forma-tion of cells with the proper shape. These results prompted us to test whether actin crosslinkers would also be involved in modulating the dynamics of actomyosin contraction during ventral furrow formation, a paradigm of tissue invagination. By screening a collec-tion of homozygous viable fly lines expressing endogenously tagged crosslinkers with fluorescent proteins (Lye *et al*, 2014), we identi-fied βH-spectrin as being upregulated at the apical surface of ventral mesodermal cells during invagination (Fig 1D, Movie EV1). Live imaging of embryos co-expressing mVenus::βH-spectrin and the plasma membrane marker GAP43::mCherry demonstrated that as mesodermal cells started constricting, βH-spectrin formed a network of medio-apical fibers, whereas in the ectoderm it remained local-ized at the junctions demarcating cell boundaries (Fig 1E–G, Movie EV2). Imaging mVenus::βH-spectrin and the myosin-II probe Sqh::mCherry showed that in mesodermal cells βH-spectrin localized to

the medio-apical actomyosin network (Fig 1H–J and cartoon in Fig 1A–C). Phalloidin staining confirmed that βH-spectrin and F-actin colocalized in medio-apical fibers (Figs 1K and L and EV1A–C). These results showing upregulation and medio-apical localization of βH-spectrin in mesodermal cells are suggestive of its potential involvement in tissue invagination. To test this hypothesis, we established a protein knockdown strategy based on a maternally expressed deGradFP-nanobody (Caussinus *et al*, 2011) to achieve efficient βH-spectrin depletion during early embryogenesis (Fig 2A). βH-spectrin depleted embryos underwent normal cellularization but displayed several defects during ventral furrow invagination as demonstrated by live imaging analysis of βH-spectrin depleted embryos expressing GAP43::mCherry. Whereas in 92% of wild-type embryos ventral furrow invagination occurred normally, only 24% of βH-spectrin depleted embryos completed invagination with no or minor abnormalities (Fig 2B–J). The remaining 76% displayed several defects including impaired apical constrictions (Fig 2K,L) and either irregular or complete failure of tissue internalization (Fig 2G–J, Movie EV3). In wild-type embryos, mesodermal cells constrict in a coordinated manner aligning their axis of contractility to the geometry of the ventral furrow primordium resulting in a shrinkage along the dorsoventral (D-V) axis and elongating along the anteroposterior (A-P) axis (Martin *et al*, 2010; Guglielmi *et al*, 2015; Chanet *et al*, 2017). In βH-spectrin knockdown embryos, cells constricted with a lower degree of A-P anisotropy compared with controls (Fig 2L) and displayed irregular shapes with some cells being more constricted than others. Together these alterations resembled the morphological abnormalities characteristic of the *twist* mutant phenotype as described in the introduction.

To test whether βH-spectrin localization in mesodermal cells is Twist-dependent (see schematics in Fig 3A and B), we generated homozygous *twist* mutant embryos in a homozygous mVenus::βH-spectrin background and followed mVenus::βH-spectrin localiza-tion using live imaging. *twist* mutant embryos formed an irregular ventral furrow, but βH-spectrin protein levels were not notably different than in control embryos (Fig 3D and E). In contrast, βH-spectrin localization changed and became predominately junctional,

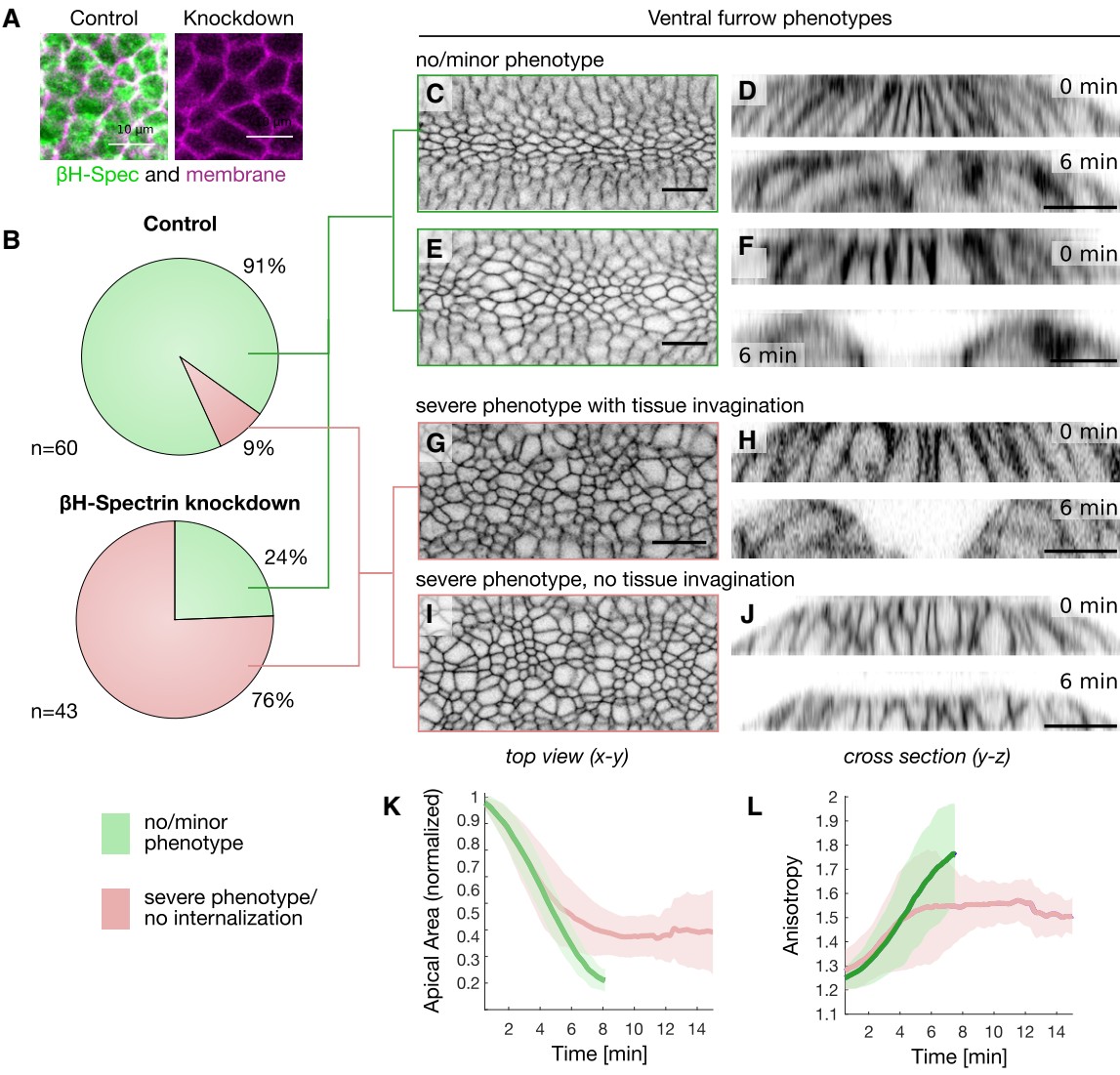

**Figure 2.** Knockdown of βH-spectrin impairs apical constriction and tissue invagination.

A *Z*-Projection of confocal image stacks spanning 5 μm of a control embryo (left) co-expressing endogenously tagged mVenus::βH-spectrin (green) and the plasma membrane marker GAP43::mCherry (magenta) and of an embryo additionally expressing a maternally driven anti-GFP nanobody KD module to knockdown βH-spectrin (right). Co-expression of the anti-GFP nanobody KD system resulted in a drastic decrease of βH-spectrin protein levels as assessed by the fluorescent signal intensity which dropped below the detection limit.

B Pie diagrams quantifying the percentage of control embryos (top, *n* = 50) and βH-spectrin knockdown embryos (bottom, *n* = 40) that formed ventral furrow either normally or with minor phenotypes (green) and the percentage of embryos that displayed severe abnormalities or did not at all undergo tissue invagination (red). The assessment of phenotypes was done in an unbiased randomized approach.

C–J Confocal images of a control embryo (C, D) and βH-spectrin knockdown embryos with a minor phenotype (E, F), with a severe phenotype that underwent tissue invagination (G, H) and with a severe phenotype that did not undergo tissue invagination (I, J). Surface view is shown in (C, E, G, I) and the cross section through the tissue at two different time points in (D, F, H, J). All embryos expressed the plasma membrane marker GAP43::mCherry. Scale bars, 20 μm.

K, L Quantification of apical surface area (K) and anisotropy (L) in control embryos (green, $n_{Control}$: six embryos) and in βH-spectrin knockdown embryos (red, $n_{βH-spectrin\ KD}$: seven embryos). Solid lines indicate the mean values and the semi-transparent regions the corresponding standard deviation.

as in lateral ectodermal cells, instead of medio-apical (Fig 3F and G and Movie EV4). In agreement with a previous study showing that βH-spectrin transcription starts at later stages of embryogenesis (Thomas & Kiehart, 1994), these results argue that Twist does not control βH-spectrin expression directly, but rather affects its localization indirectly. Because Twist regulates the spatiotemporal organization of Rho signaling (Mason *et al*, 2013) (see Fig 3A and B for

a schematic), we tested whether Rho activation controls βH-spectrin dynamics and localization. We employed optogenetics to stimulate RhoGEF2 and induce an acute burst of Rho signaling (Izquierdo *et al*, 2018) (see schematic in Fig 3C) on the dorsal surface of the embryo in ectodermal cells where Twist is not expressed and βH-spectrin is localized to the junctions (Fig 3H). Spatially confined optogenetic activation using two-photon illumination (Guglielmi &

De Renzis, 2017) caused both βH-spectrin medio-apical localization and a ~ 1.7-fold increase in its overall levels at the apical surface (Fig 3H–R), thus demonstrating that Rho signaling is sufficient to induce βH-spectrin apical upregulation and medio-apical assembly. Injection of the Rock inhibitor Y-27632 before optogenetic activation did not result in βH-spectrin or myosin-II upregulation. Thus,

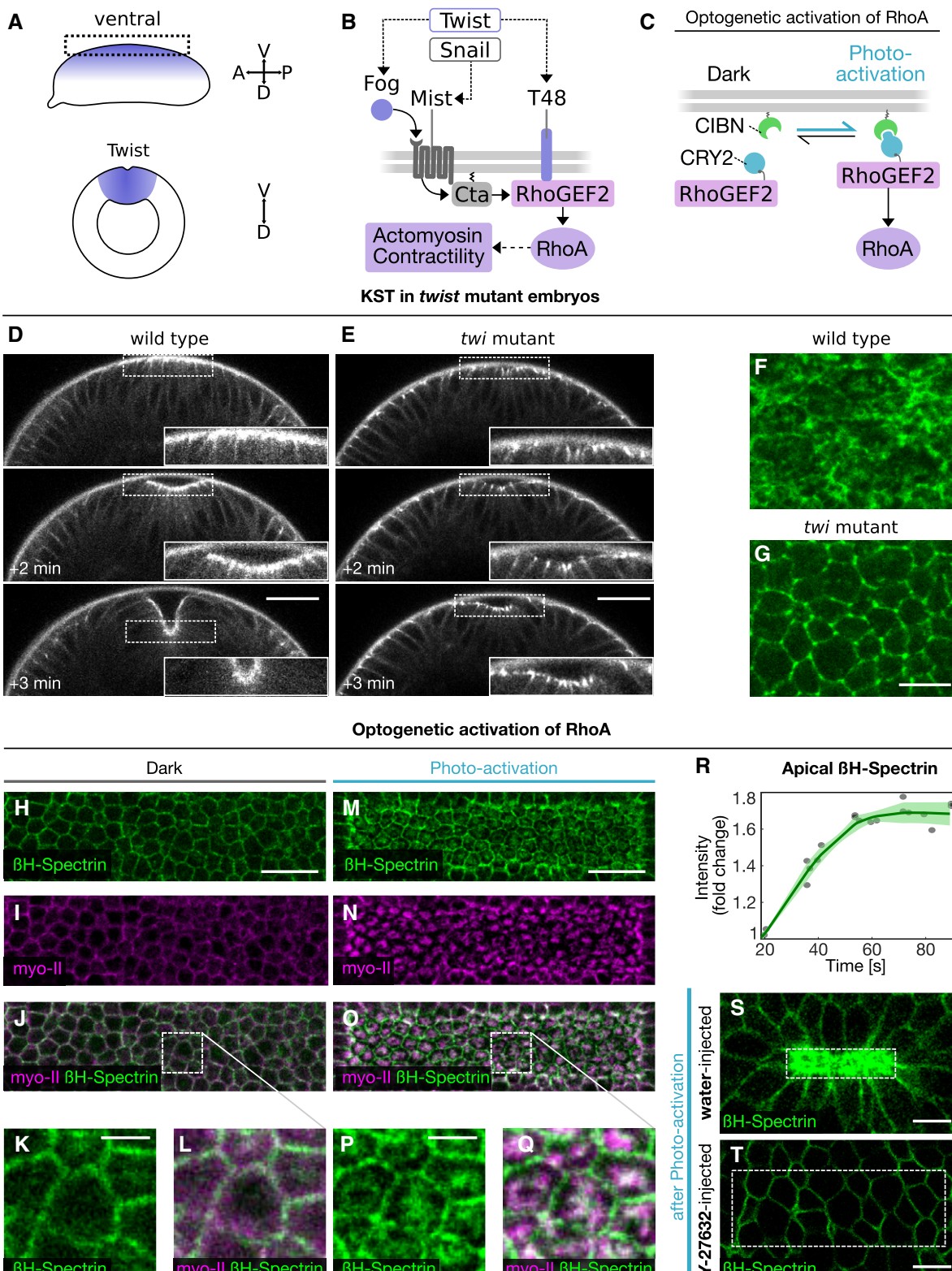

**Figure 3.**

**Figure 3.  Medio-apical βH-spectrin localization is controlled by the mesoderm-specific transcription factor Twist and Rho signaling activation.**

A       Schematic showing the expression domain of the transcription factor Twist on the ventral side of the *Drosophila* embryo along the anterior-posterior axis (top: lateral view; bottom: cross section).

B       Schematic showing the genetic network that induces actomyosin contractility at the apical surface of mesodermal cells. The transcription factors Twist and Snail regulate expression of the G-protein-coupled receptor (GPCR) Mist, its ligand Fog and the Gα Cta, as well as T48, which recruits the RhoA activating factor RhoGEF2 to the apical surface. Fog-mediated activation of Mist triggers RhoGEF2 activity through Cta and thus induction of actomyosin contractility.

C       Schematic of the optogenetic system employed to activate Rho signaling. RhoGEF2-CRY2 is cytosolic in the dark and interacts upon photo-activation with blue light with its binding partner CIBN which is anchored at the plasma membrane.

D, E    Still frames of a movie showing endogenously tagged mVenus::βH-spectrin in the cross section of a gastrulating wild-type (D) and *twist* mutant (E) embryo imaged using two-photon microscopy. The top frame corresponds to the time point where an initial bending of the apical surface was observed, the middle frame was taken 2 min and the lower frame 3 min afterward. Scale bars, 30 μm.

F, G    Confocal images showing apical βH-spectrin (endogenously tagged mVenus::βH-spectrin) in a wild-type (F) and *twist* mutant (G) embryo. Scale bars, 20 μm. Medio-apical βH-spectrin fibers observed in wild-type embryos were absent in *twist* mutant embryos.

H–Q     Confocal or two-photon microscopy images of the apical surface of an embryo mounted with the dorsal side facing the objective and co-expressing endogenously tagged mVenus::βH-spectrin (green, H, K, M, P), the myosin-II marker Sqh::mCherry (magenta, I, N), and the optogenetic modules as explained in (C). The apical surface is shown before photo-activation (H–L) and 1 min after photo-activation (M–Q). Panels (J, O) show a merge of βH-spectrin and myosin-II and the dashed rectangle indicates regions shown in (L and Q) at higher magnification. Panels (K, L) and (P, Q) are zoom-ins into the region indicated in (J) an (O), respectively. Scale bars, 20 μm (H–J, M–O), 2.5 μm (K, L, P, Q).

R       Quantification of apical βH-spectrin upon optogenetic stimulation of Rho signaling. The graph shows the mean intensity of βH-spectrin (green) at the apical surface over time as the fold change relative to the initial time point (before photo-activation). The semi-transparent region indicates the standard deviation, and dots indicate individual data points (*n* = 5 embryos).

S, T    Two-photon microscopy images of apical βH-spectrin (green) 1.5 min after optogenetic stimulation of Rho signaling in embryos that were previously injected with water (S) or the ROCK inhibitor Y-27634 (T). Dashed boxes indicate photo-activated cells. Y-27634-injected embryos did not undergo apical constriction and did not accumulate βH-spectrin upon photo-activation, while water-injected embryos constricted and showed increased βH-spectrin levels. Scale bars: 10 μm.

medio-apical βH-spectrin upregulation is dependent on the Rock-myosin-II branch of the Rho signaling system (Figs 3S and T and EV2A–I), consistent with the role of Rock in organizing the medio-apical actomyosin network (Mason *et al*, 2013).

These results, showing Twist and Rho control over βH-spectrin medio-apical localization, prompted us to test whether the impaired contractile behavior caused by βH-spectrin knockdown results from impaired pulsatile behavior rather than lack of apical constriction *per se*. We followed myosin-II (Sqh::mCherry) dynamics in embryos expressing the junctional protein E-cad::mNeonGreen demarcating the cell boundaries using high-temporal resolution confocal imaging (Movie EV5). In βH-spectrin knockdown embryos, myosin-II displayed normal medio-apical accumulation (Movie EV5). However, over time the apical actomyosin network broke in cells that enlarged abnormally resulting in tissue tears (Fig 4A–C) and local disruption of adherens junctions (Fig EV3A–F). A similar phenotype was observed in βH-spectrin knockdown embryos expressing the F-actin reporter LifeAct::mNeonGreen (Movie EV6), suggesting that βH-spectrin is not required for initial assembly of the medio-apical actomyosin network but rather for maintaining its structural integrity as cells constrict. Quantitative analysis of mVenus::βH-spectrin signal over the course of apical constriction reveals that its levels progressively increased during the constriction phase and remained constant during relaxation (Fig EV4A–F). By inspecting the contractile behavior of βH-spectrin knockdown cells, we noticed that while some cells constricted in a ratcheted manner maintaining the constricted states also in the interval between subsequent myosin-II pulses, some others constricted in a non-ratcheted manner (Fig 4D and E). Quantification of the contractile behavior in more than 500 individual cells revealed that in control embryo cells constricted in a coordinated manner over time in a such a way that after 7 min from the onset of ventral furrow formation most of the cells (91%) shrank their apical area to < 50% of the initial value. In contrast, βH-spectrin knockdown caused loss of coordination with as many as 44% of the cells displaying impaired constriction (i.e., apical area > 50% and < 100% of initial values) and 6% of the cells expanding

(Fig 4F–J). While in control embryos cells displayed positive constriction rate values throughout ventral furrow invagination, in βH-spectrin knockdowns cells with impaired contractions displayed both positive and negative rates indicating alternating cycles of surface contraction and expansion (Fig EV5A and B). Quantification of the constriction and expansion rates show that the constriction rate was reduced by ~ 24%, whereas the expansion rate increased by ~ 60% on average (Figs 4K and EV5C and D), demonstrating that the change in the expansion rate is 2.5 times higher than the change in the constriction rate. Taken together, these results are consistent with a role of βH-spectrin in apical ratcheting.

In conclusion, the results presented in this study provide new insights into the mechanisms controlling apical ratcheting during ventral furrow invagination, linking the activity of the mesoderm-specific transcription factor Twist and of Rho signaling to βH-spectrin medio-apical localization and function. Previously, Twist has been shown to be required for apical ratcheting but how ratcheting is controlled at the molecular level was unknown. Our results suggest that actin cross-linking plays a critical role in this process. βH-spectrin belongs to the spectrin protein family comprising long heterodimeric proteins consisting of α- and β-subunits, which dimerize and assemble into tetramers. These tetramers constitute the basic actin cross-linking unit, where actin binding is conferred by the β subunit at each end, and form networks that interconnect actin filaments at the plasma membrane (Bennett & Baines, 2001; Stabach *et al*, 2009). In *Drosophila* epithelial cells, the spectrin cytoskeleton is polarized with α- and β_H-dimers enriched at the apical domain and α- and β-dimers at the basal surface (Stabach *et al*, 2009). Our results are consistent with this polarized distribution and with the described role of βH-spectrin in controlling follicle cell morphogenesis during *Drosophila* oogenesis, but argue that βH-spectrin is not required for apical constriction *per se* as suggested based on analysis of fixed samples (Stabach *et al*, 2009). The function of βH-spectrin during apical constriction might be to cross-link and stabilize the medio-apical actomyosin network and/or to ensure its attachment to the junctions so that cells can maintain their

constricted state also when neighboring cells constrict and exert forces on them. This regulatory mode further highlights the importance of actin crosslinkers in tissue mechanics and morphogenesis. Given the role of βH-spectrin in mechanosensation (Johnson *et al*, 2007) and regulating of YAP localization both in *Drosophila* and human cells (Fletcher *et al*, 2015; Wong *et al*, 2015), it will be interesting to test the impact of apical ratcheting and dynamic control of apical constriction on downstream signaling events and tissue differentiation programs. Future work should also focus on how regulation of actin cross-linking interplays with other cellular processes

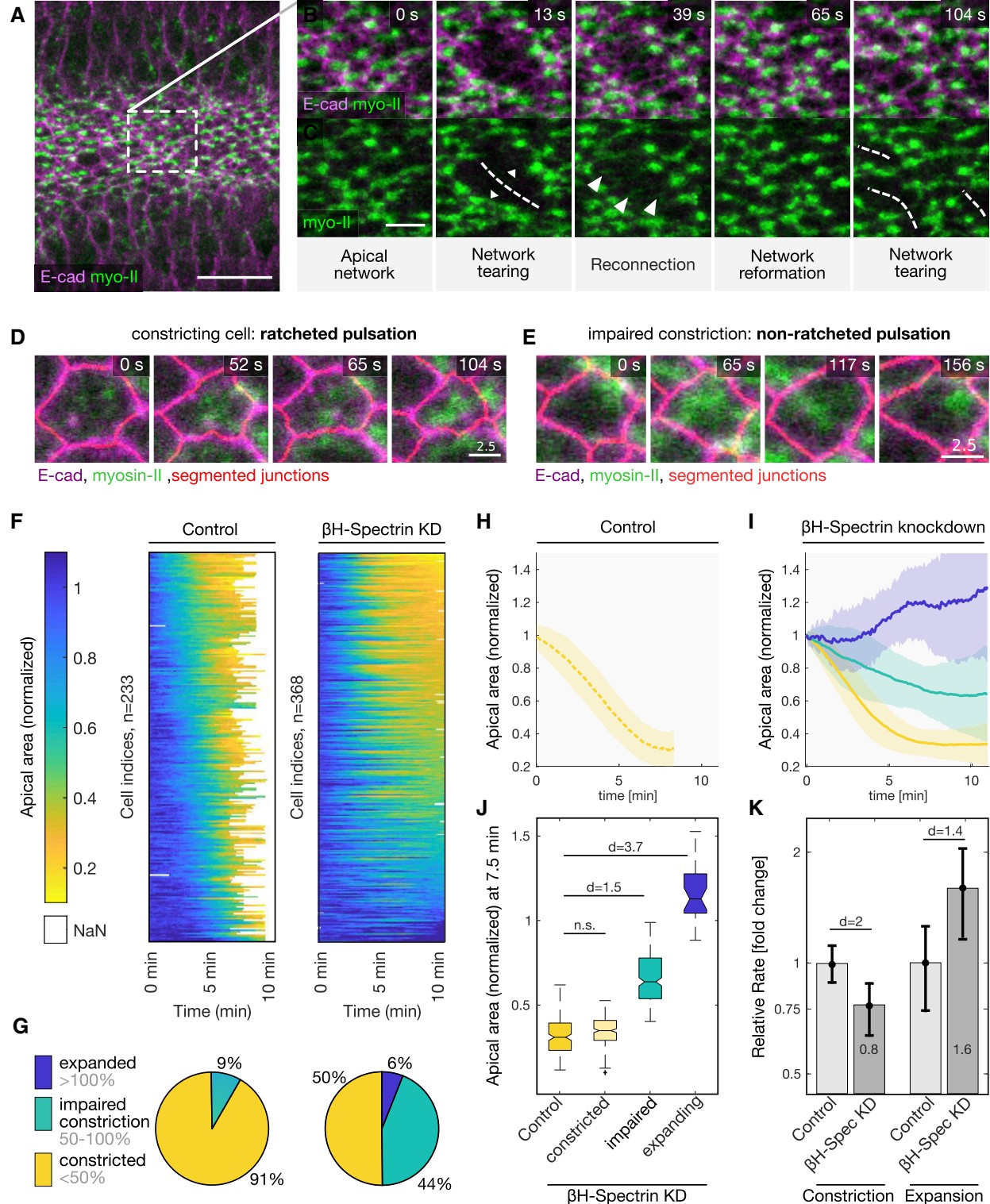

**Figure 4.**

◀

**Figure 4. βH-spectrin is required for ratcheting pulsatile apical constrictions.**

A–C   Confocal images of the apical cell surface of a βH-spectrin knockdown embryo showing the myosin-II marker Sqh::mCherry (green) and E-cadherin::mNeonGreen (magenta) during ventral furrow formation. (A) myosin-II accumulates normally at the apical surface and cells undergo an initial phase of constriction. The dashed rectangle indicates the region shown in (B and C) at higher magnification. Scale bar, 30 μm. Panels (B, C) show a sub-region within the ventral tissue at different time points. Initially, a supracellular myosin-II network formed (0 s), before intercellular connections broke (13 s) resulting in membrane tears (dashed line) with tissue retractions (small arrow heads) along the anterior-posterior axis. The intercellular network re-established (39, 65 s) as new actomyosin fibers formed (big arrow heads) to reconnect neighboring cells. The intercellular network broke again causing tears (dashed lines) at different positions (104 s). Scale bars, 5 μm.

D, E   Confocal images showing the myosin-II marker Sqh::mCherry (green), E-cadherin::mNeonGreen (magenta) and the segmented membrane (red) of a single cell at different time points in a control embryo undergoing ratcheted constriction (D) and in a βH-spectrin knockdown embryo undergoing non-ratcheted pulsations (E). Scale bars, 2.5 μm.

F   Heatmaps showing the apical surface of individual cells over time of control embryos (left) and βH-spectrin knockdown embryos (right) normalized to the initial time point ($n_{Control}$ = 233; $n_{βH-spectrinKD}$ = 368; of five different embryos each). The color-coded scale bar is depicted on the left. Dark blue indicates a cell area bigger than the initial value (> 1), and yellow indicates an area constricted to less than 1/5 of the initial area (< 0.2). Three different cell behaviors in βH-spectrin knockdown embryos were identified: cells that constricted, cells that showed impaired constriction, and cells that expanded.

G   Quantification of the percentage of cells that constricted (to an area < 50%, yellow), cells that showed impaired constriction (to an area > 50 and < 100%, green), and cells that expanded (to an area > 100%, blue) after 7 min in control embryos and in βH-spectrin knockdown embryos.

H, I   Graph showing the average area over time of control cells (H, $n$ = 235 cells) and cells in βH-spectrin knockdown embryos (I) that constricted (yellow, $n$ = 274), of cells that showed impaired constriction (green, $n$ = 232), and of cells that expanded (blue, $n$ = 32). The lines indicate the mean value and the semi-transparent region the standard deviation.

J   Boxplots showing the distribution of apical cell area values 7.5 min after the onset of ventral furrow formation of ventral cells in control embryos, of ventral cells in βH-spectrin knockdown embryos that constricted, and ventral cells in βH-spectrin knockdown embryos that displayed impaired constriction and expanded. Sample numbers are equal to I. The values are presented as percentage of initial cell area. ANOVA result (all data points): $F_{(3,717)}$ = 763.9, $P$ = 9.6e-223; Cohen's $d$ compared with the control: $d_{constricted}$ = 0.1, $d_{impaired}$ = 1.5, $d_{expanded}$ = 3.7. Due to a big effect size (high sample number), statistical significance was assessed based on Cohen's $d$. A Cohen's $d$ < 0.5 was considered not significant (n.s.).

K   Quantification of the peak constriction rate and expansion rate relative to control cells. Control cells and cells in βH-spectrin knockdown embryos that showed impaired constriction or expanded were compared. The mean value and standard deviation are shown. Please note the logarithmic scale. While in βH-spectrin knockdown cells the constriction rate was ~ 24% lower (mean value of 0.8) compared with control cells, the expansion rate increased to ~ 60% (mean value of 1.6). The absolute change of the expansion rate was 2.5 times higher than the change of the constriction rate. ANOVA result (all data points): $F_{(3,717)}$ = 349.2, $P$ = 9.7e-140; pairwise comparison of βH-spectrin knockdown cells to control using Cohen's $d$ > 1.4.

Data information: (J, K) Sample numbers are the same as in (H, I). Boxplots: The central mark, the bottom, and the top edge of each box plot indicate the median, the 25th and 75th percentiles, respectively. Whiskers extend to the most extreme data point and crosses indicate outliers. Notches indicate comparison intervals.

---

involved in apical ratcheting, such as membrane turnover (Miao *et al*, 2019).

# Material and Methods

## Live imaging and optogenetics

Adult flies were kept at 18°C in cages with removable agar plates at the bottom. Stage 5 (cellularizing) embryos were selected under halocarbon oil using a standard stereomicroscope. Embryos were dechorionated in 100% sodium hypochlorite for 2 min, washed in distilled water and mounted onto 35-mm glass-bottom dish (MatTek) in PBS. Embryos were mounted with the ventral surface facing the objective unless stated otherwise. Live imaging was performed at 20°C using a Zeiss LSM780 NLO confocal microscope (Carl Zeiss) equipped with a tunable (690–1,040 nm) 140-fs pulsed multi-photon laser (Chameleon; Coherent) with a repetition rate of 80 MHz and a 40× C-Apochromat (NA 1.20) water immersion objective (Carl Zeiss). The microscope was operated using Zen Black 2012 (Carl Zeiss) software and Pipeline Constructor Macro (Politi *et al*, 2018). GFP and mVenus live reporters were imaged either with a 488 nm multiline Argon laser (Lasos Laser GmbH) in confocal mode and the internal spectral GaAsP detector or with 950 nm multi-photon excitation and the PMT-NDD detectors in the non-descanned mode, as specified in the corresponding figure legends. mNeonGreen and mCherry probes were imaged in confocal mode using an Ar laser at 488 nm and a HeNe laser at 561 nm, respectively. Two-channel confocal imaging was done using two separate channel tracks (non-simultaneous

excitation of the fluorescent probes) to prevent signal bleed-through.

For optogenetic experiments, flies were kept in the dark and samples were prepared using a red light–emitting LED lamp, to prevent photo-activation. While orienting the sample at the microscope, a deep amber lighting filter (Cabledelight) was used to block bright-field illumination. To image mVenus::βH-spectrin upon Rho signaling optogenetic activation, embryos were mounted with the dorsal side facing the objective. A pre-activation image stack at the apical surface was acquired using 561 nm laser excitation to visualize Sqh::mCherry or GAP43::mCherry. Based on the pre-activation stack, a region of interest (ROI) was defined in which the apical surface was photo-activated and at the same time the mVenus::βH-spectrin signal was recorded using multi-photon illumination. A 6.6 μm deep focal volume at the apical surface was photo-activated in a bi-directional scanning mode with a total dwell time of 1.27 μs, a pixel size of 0.21 μm, and using a laser power of 11.5 mW. The mCherry signal was recorded after photo-activation at repetitive cycles. To test co-localization with myosin-II, the recorded mCherry stack and the photo-activated region in which also mVenus::βH-spectrin signal was recorded had the same dimensions and positioning. To measure medial and junctional βH-spectrin upon optogenetic activation, the mCherry signal was recorded in three focal planes (spanning 2 μm) centered at the middle plane of the photo-activation stack.

## Embryo injection optogenetic experiment

Embryos expressing the Rho-activating optogenetic system were dechorionated with sodium hypochlorite (100%), aligned on a glass

coverslip coated with heptane glue and covered with a layer of halo-carbon oil Voltalef 10S (Merk). The coverslip was mounted on a microscope slide platform to visualize embryos on a standard upright microscope using a 10× objective (Carl Zeiss). Microinjection was performed using an Eppendorf microinjector (model 5242). Microinjection pipettes were pulled from borosilicate glass capillaries (1.2-mm outer diameter × 0.94-mm inner diameter; Harvard Apparatus), using a P-97 Flamming/brown puller (Sutter Instrument). Embryos were injected in the posterior pole with water (control) or the ROCK inhibitor Y-27632 at a concentration of 50 mM (in aqueous solution). The entire sample preparation and injection procedure were done in red light to prevent pre-mature activation of the optogenetic system.

Optogenetic experiments following embryo injections were done as described above at a Zeiss LSM780 NLO microscope, except that the cells were simulated at the apical surface with a single photo-activation using a laser power of 13 mW. The mCherry signal was recorded for 1.5 min, and a final 2-photon stack with lower zoom was recorded.

**Image and data analysis**

Images were processed and analyzed using Fiji (https://fiji.sc/) and MATLAB-R2017b (MathWorks). The data and meta data were extracted from the acquired images using the MATLAB Bioformats toolbox (http://www.openmicroscopy.org). To analyze and categorize βH-spectrin knockdown phenotypes, multi-temporal image stacks of control and βH-spectrin knockdown embryos were automatically presented to the experimentalist in a randomized order and under blinded experimental condition. Upon visual inspection, the process of ventral furrow formation for each sample was scored according to the following categories: "normal", "minor phenotype", "severe phenotype with tissue invagination", and "severe phenotype without tissue invagination". To quantify apical surface dynamics, individual cells were segmented from apical sections of the plasma membrane (based on the GAP43::mCherry signal) and basic geometric cell features including cell area and anisotropy were extracted using the Embryo Development Geometry Explorer (EDGE; https://github.com/mgelbart/embryo-development-geometry-explorer) package. Gaps smaller than three time points resulting from unsuccessful segmentation were linearly interpolated. The cell data were averaged per embryo over time and the mean (with standard deviation) of multiple embryos plotted in Fig 2. In Fig 4, the cell data of multiple embryos were pooled to analyze individual and group-specific behaviors. To analyze cumulative group-specific cell behaviors, the constriction per cell after 7 min relative to the initial time point was calculated and divided into three categories (< 0.5; 0.5 < 1; > 1) and plotted. Cells that were tractable for only one standard deviation less than the median were excluded from the heat maps. The apical area depicted in the heat maps and in the graphs were smoothened with a window of three time points corresponding to 11 s and ~ 20 s, respectively. The constriction rate was calculated by dividing the apical area by the time interval and smoothened with a window of 10 time points (~ 38 s). To analyze cell ratcheting, the values of the surface area for each cell at a given time point was subtracted by the trend to normalize the data. The trend was estimated by fitting a $3^{rd}$ degree polynomial function to the surface area values over time. For quantifying pulsation, both local maxima and minima (inverted data) were identified from the normalized surface area using the MATLAB *findpeaks* function. The mean difference in area between consecutive maxima or minima was calculated and averaged to get a single value for each cell. This value describes the mean amplitude of the changes in area upon pulsation and thus ratcheting.

Similarly, pulsation peaks (constriction and expansion peaks) were identified using the *findpeaks* function applied to the unsmoothed constriction rate. The expansion rate was analyzed by inverting the constriction rate. Only positive pulsation peaks were considered (ignoring non-positive local maxima). For individual cells, pulsation peaks were identified and the mean rate per cell was calculated. Cells in βH-spectrin KD embryos were categorized as described above and plotted as boxplots. To compare the change in constriction versus change in expansion rate upon βH-spectrin KD, cells with impaired constriction and cells that expanded in βH-spectrin KD embryos were grouped and compared with the control. The rates of control and βH-spectrin KD cells were normalized to the mean value of the control group resulting in a fold change of the constriction and expansion rate.

To calculate βH-spectrin levels upon optogenetic activation of Rho signaling, the mean intensity of the mVenus::βH-spectrin signal within a 6.6 μm-spanning image stack was measured at each time point after photo-activation. A background signal was estimated as the mean signal intensity at the lowest focal plane. After subtraction of the background from the measured signal, the intensity values were normalized relative to the initial time point. Due to minimal variations in the time interval, the intensity values were linearly interpolated and based on that, the mean was calculated and plotted with the corresponding standard deviation.

To quantify apical mVenus::βH-spectrin signal per cell, embryos expressing mVenus::βH-spectrin and GAP43::mCherry were co-imaged, and an image stack spanning ~ 4 μm from the most apical confocal planes was analyzed. A background signal was estimated as one standard deviation subtracted from the mean of the image stack and the background was subtracted. The image stack was projected in 2D calculating the mean intensity and imported into EDGE together with the co-acquired GAP43::mCherry signal. Cells were segmented based on the GAP43::mCherry signal, and βH-spectrin levels were analyzed using the in-build function in EDGE. Average curves of area and βH-spectrin levels containing data of multiple samples were analyzed using customized MATLAB scripts. The constriction rate was calculated by dividing area values by the time interval, and the time points of pulsation peaks were identified using the MATLAB *findpeaks* function. The expansion rate was calculated likewise using the inverted constriction rate. Within a time window spanning 50 s, centered at the time point of an isolated peak, the apical area, constriction rate, and βH-spectrin level were identified. The data for all peaks were overlaid and analyzed by calculating the mean and standard deviation.

Co-localization was analyzed using the Fiji plugin "Coloc 2" according to the online documentation (https://imagej.net/Coloc_2).

Surface projection was done by splitting the image stack into a $(x/y)$ grid and analyzing the $z$-intensity profile within each unit to find the surface intensity peak. Each grid unit was associated with the identified $z$-slices and a gauss-filter applied. Per grid unit, the relevant confocal sections were projected proportionally. To prevent edge effects, the original image stack was expanded in $x$–$y$

dimension by half a grid size and the same procedure was repeated. To do so, the expanded boundaries of each image were filled by the mean intensities of the outer margin (spanning half a grid size). The final surface projection resulted from the mean of the projections of the original and expanded stack. The βH-spectrin signal served as a reference for the positional information of the phalloidin staining; thus, identical confocal image planes were projected for the phalloidin and βH-spectrin channel.

## Statistical analysis

Statistical significance of data pairs was tested by two-sample Student's $t$-tests and calculating the $P$-value. To compare multiple samples, one-way analysis of variance (ANOVA) was performed followed by a *post hoc* Tukey's test. Due to high sample numbers of the presented date, significance between two sample populations of a dataset was additionally determined by analyzing the effect size by means of Cohen's $d$:

$$d = \sqrt{\frac{(\bar{x_1} - \bar{x_2})^2}{s}}$$

where $\bar{x_1} - \bar{x_2}$ is the difference between the sample pair's mean values and $s$ is the complete dataset's maximal standard deviation. Sample populations with Cohen's $d$ values of $d < 0.5$ indicate low effect size and were thus considered as not significantly different.

## Immunostaining

Flies were kept at 18°C for 8 h in cages with a removable agar plate at the bottom. The agar plate was removed, covered with halocarbon oil, and embryos at the late stage of cellularization were selected using a standard stereoscope. Embryos were dechorionated in 100% sodium hypochlorite for 2 min and fixed in a mixture of equal amounts of 100% heptane and 4% paraformaldehyde/PBS (Electron Microscopy Sciences) for 20 min. Devitellinized embryos were washed in 100% methanol, permeabilized in PBT (PBS, 1% BSA, and 0.05% Triton X-100) and incubated in blocking solution (PBS, 6% BSA, and 0.05% Triton X-100) for 1 h at room temperature. For mVenus::βH-spectrin staining, embryos were incubated in a PBST staining solution containing FluoTag®-X4 single-domain anti-GFP antibody conjugated with Abberior®Star635P fluorophore (NanoTag Biotechnologies GmbH, Cat no: N0304-Ab635P-S) at a dilution of 1:250 for a final fluorophore concentration of 20 nM for 1.5 h at room temperature and washed in PBST. Embryos were mounted on a 0.16–0.19-mm-thick (thickness 1.5) cover glass (Glaswarenfabrik Karl Hecht GmbH & Co KG) in ProLong Gold Antifade Mountant (Molecular Probes/Thermo Fisher Scientific).

## Actin staining

Embryos at the end of the cellularization stage were collected and dechorionated as described above, fixed in formaldehyde-saturated heptane for 40 min and placed onto two-sided sticky tape covered with PBS. The vitelline membrane was removed using forceps under a standard stereomicroscope. Embryos were collected in PBS, 0.1% Triton X-100 solution. Devitellinized embryos were washed in PBS, 1% BSA, 0.05% Triton X-100 solution and incubated in PBS, 6%

BSA, 0.05% Triton X-100 blocking solution for 1 h at room temperature. Embryos were stained using FluoTag®-X4 as described above for 1.5 h at room temperature before they were incubated in actin staining solution (7.5 µl of phalloidin-atto647N [Sigma-Aldrich] stock solution [20 µM in methanol] in a volume of 1 ml of PBS, 0.1% Triton X-100) for 1.5 h at room temperature. The embryos were washed in PBS, 1% BSA, 0.05% Triton X-100 solution and embedded in low-melting agarose (0.4%, w/v) on a MatTek glass-bottom dish with the ventral surface facing the glass bottom.

## STED microscopy

Stimulated emission depletion (STED) was performed on a Leica SP8 STED 3× microscope (DMI6000) equipped with white-light laser (470–670 nm) and 592, 660 and 775 nm STED lasers. Abberior®Star635P–stained samples were imaged using 633-nm excitation in combination with the 775-nm STED laser. A 100× oil HC PL APO CS2 objective (NA 1.40) with type F immersion liquid was used at 22.5°C. The microscope was operated using LAS X (Leica) software. Emission light was detected in a spectral range between 640 and 750 nm using gated detection on Leica HyD detectors. Images were acquired with a pixel size of 15 nm, an averaging of 16, and a dwell time of 1.2 µs, and the pinhole opening was set to 0.93 A.U.

## Anti-GFP nanobody-mediated protein knockdown

Female flies homozygous for mVenus::βH-spectrin were crossed to heterozygous males expressing the anti-GFP-nanobody under the control of the maternal tubulin promoter, to generate flies homozygous for mVenus::βH-spectrin and co-expressing Sqh::mCh and the anti-GFP nanobody.

## Fly genetics

Fly crosses were kept at 22–25°C, and the following fly lines were generated by standard procedures.

To co-visualize actomyosin and the plasma membrane:
w[*];P[w+, sqhp>GAP43::mCherry]; P[w+, sqhp>sqh::GFP].

To visualize the plasma membrane upon βH-spectrin protein KD:
w[*]; M[w+, UASp>Nslmb.vhhGFP4]/P[w+, sqhp>GAP43::mCherry]; kst[w+, CPTI002266], mat.tubulin >Gal4::VP16/kst[w+, CPTI002266]. A fly line having the balancer chromosome CyO instead of the anti-GFP nanobody-containing chromosome served as control.

To visualize βH-spectrin in *twist* mutant embryos:
w[*];twi*ey53*, *Δhalo*/CyO; kst[w+, CPTI002266]. *twist* mutant embryos were identified by the *halo* phenotype during cellularization.

To visualize myosin-II upon protein KD of βH-spectrin:
w[*]; M[w+, UASp>Nslmb.vhhGFP4]/P[w+, sqhp>sqh::mCherry]; kst[w+, kst[w+, CPTI002266], mat.tubulin >Gal4::VP16/kst[w+, CPTI002266].

To visualize both myosin-II and E-cadherin upon βH-spectrin protein KD:
w[*]; M[w+, UASp>Nslmb.vhhGFP4]/P[w+, sqhp>sqh::mCherry]; kst[w+, kst[w+, CPTI002266], mat.tubulin >Gal4::VP16/kst[w+, CPTI002266], P[w+, ubip>shg::mNeonGreen]. A fly line having the balancer chromosome CyO instead of the anti-GFP nanobody-containing chromosome served as control.

To visualize both F-actin and the plasma membrane upon βH-spectrin protein KD:

w[*]/P[w+, sqhp>GAP43::mCherry]; M[w+, UASp>Nslmb.vhh GFP4]/P[w+, sqhp>LifeAct::mNeonGreen]; kst[w+, kst[w+, CPTI 002266], mat.tubulin >Gal4::VP16/kst[w+, CPTI002266]. A fly line having the balancer chromosome CyO instead of the anti-GFP nano-body-containing chromosome served as control.

For optogenetic experiments and visualization of βH-spectrin, the following females were generated, crossed to wild-type males and maintained in the dark:

To visualize βH-spectrin and myosin-II: w[*], P[w+, UASp>-CIBNa]; P[w+, Sqh::mCherry]/P[w+, UASp>RhoGEF2-CRY2]; kst[w+, CPTI002266], P[w+, Osk>Gal4::VP16]/kst[w+, CPTI002266].

To visualize βH-spectrin and the plasma membrane: w[*]/P[w+, UASp>CIBN]; P[w+, GAP43::mCherry]/P[w+, UASp>RhoGEF2-CRY2]; kst[w+, CPTI002266], P[w+, Osk>Gal4::VP16]/kst[w+, CPTI 002266].

## Fly stocks

w[*]; P[w+, sqhp>sqh::mCherry]/CyO; Dr/TM3, Ser, Sb. Myosin regulatory light chain (spaghetti squash) tagged with the fluorescent protein mCherry expressed under the spaghetti squash (*sqh*) promoter

w[*];If/CyO; P[w+, sqhp>sqh::GFP]/TM3, Ser. Myosin regulatory light chain (spaghetti squash) tagged with the fluorescent protein GFP expressed under the spaghetti squash (*sqh*) promoter

w[1118];+; kst[w+, CPTI002266]. βH-spectrin (kst, karst) tagged with mVenus at the endogenous level, generated by the Cambridge Protein Trap Insertion (CPTI) project and obtained from the Kyoto *Drosophila* Genomics and Genetic Resources stock center (DGRC stock number: 115285).

w1118; *twi^ey53^*, *Δhalo*/Cyo. A fly line carrying the *twist* mutant allele *twi^ey53^* combined to Δhalo (Df(2L) dpp[s7-dp35] 21F1–3; 22F1–2).

w[*], P[w+, UASp>CIBN]. *UAS-Gal4-driven* CIB1 N-terminal domain (CIBN) fused a CAAX box (membrane anchor).

w[*]; P[w+, UASp>RhoGEF2-CRY2]/CyO. *UAS-Gal4-driven* DHPH domain of *Drosophila* RhoGEF2 fused to the photosensitive PHR domain of CRY2.

yw[*]; P[w+, sqhp>Gap43::mCherry]/CyO. The fluorescent protein mCherry tagged with a GAP43 membrane anchor expressed under the *sqh* promoter.

yw[*]; M[w+, UASp>Nslmb.vhhGFP4]/CyO. *UAS-Gal4-driven* anti-GFP-nanobody fused to the F-Box protein Slimb (Bloomington stock number: 58740).

w[*]; +; P[w+, mat.tubulin>Gal4::VP16]. Maternally deposited transcription factor Gal4 expressed under the maternal tubulin (mat.tubulin) promoter (Bloomington stock number 7063).

New fly lines generated in this study:

w[*]; If/CyO; P[w+, ubip>shg::mNeonGreen]. E-Cadherin (shot-gun; shg) tagged with the fluorescent protein mNeonGreen under the *ubiquitin* (*ubi*) promoter (generated by P-element transformation).

w[*]; P[w+, sqhp>LifeAct::mNeonGreen]; MKRS/TM6. The F-actin probe LifeAct tagged with the fluorescent protein mNeonGreen under the *sqh* promoter (generated by P-element transformation).

## Data availability

No data were deposited in a public database.

**Expanded View** for this article is available online.

## Acknowledgements

We thank all members of the De Renzis laboratory for helpful discussion and J. Hartmann and A. Runge for critical reading of the manuscript. We thank Alessandra Reversi and the EMBL *Drosophila* Injection Service for helping with embryo injection and the advanced light microscopy facility (ALMF) for their advice and assistance. This work was supported by EMBL internal funding available to S.D.R. We thank the Bloomington *Drosophila* Stock Center and the *Drosophila* Genomics Resource Center for providing fly stocks.

## Author contributions

Experiment conception and design: DK, TM, CPC, SDR; Data collection: DK, CPC, TM (Fig 3H–R); Writing manuscript: DK and SDR.

## Conflict of interest

The authors declare that they have no conflict of interest.

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
