## [Review Process File · EMBO Reports]

β H-spectrin is required for ratcheting apical pulsatile constrictions during tissue invagination

Stefano De Renzis, Daniel Krueger, Cristina Cartes, and Thijs Makaske
DOI: [10.15252/embr.201949858](https://doi.org/10.15252/embr.201949858)

Corresponding author(s): Stefano De Renzis (derenzis@embl.de), Daniel Krueger (daniel.krueger@embl.de)

Review Timeline:	Submission Date:	10th Dec 19
	Editorial Decision:	21st Jan 20
	Revision Received:	1st Apr 20
	Editorial Decision:	5th May 20
	Revision Received:	13th May 20
	Accepted:	20th May 20

Editor: Deniz Senyilmaz Tiebe

Transaction Report:

Dear Stefano,

Thank you for submitting your manuscript to our journal. We have now received two referee reports, which are copied below.

Please accept my apologies for this unusual delay in getting back to you, as I mentioned before, it took longer than usual to receive the full set of referee reports due to the recent holiday season.

As you can see, the referees express interest in the analysis. However, they also raise a number of concerns that need to be addressed to consider publication here. I will not repeat the individual points raised, but I find the reports informed and constructive, and believe that addressing the concerns raised will significantly strengthen the manuscript.

Given these constructive comments, we would like to invite you to revise your manuscript with the understanding that the referee concerns (as in their reports) must be fully addressed and their suggestions taken on board. Please address all referee concerns in a complete point-by-point response. Acceptance of the manuscript will depend on a positive outcome of a second round of review. It is EMBO reports policy to allow a single round of revision only and acceptance or rejection of the manuscript will therefore depend on the completeness of your responses included in the next, final version of the manuscript.

1. A data availability section providing access to data deposited in public databases is missing (where applicable).
2. Your manuscript contains statistics and error bars based on $n=2$ or on technical replicates. Please use scatter plots in these cases.

Supplementary/additional data: The Expanded View format, which will be displayed in the main HTML of the paper in a collapsible format, has replaced the Supplementary information. You can submit up to 5 images as Expanded View. Please follow the nomenclature Figure EV1, Figure EV2 etc. The figure legend for these should be included in the main manuscript document file in a section called Expanded View Figure Legends after the main Figure Legends section. Additional Supplementary material should be supplied as a single pdf labeled Appendix. The Appendix includes a table of content on the first page with page numbers, all figures and their legends. Please follow the nomenclature Appendix Figure Sx throughout the text and also label the figures according to this nomenclature. For more details please refer to our guide to authors.

2) individual production quality figure files as .eps, .tif, .jpg (one file per figure).

3) a .docx formatted letter INCLUDING the reviewers' reports and your detailed point-by-point responses to their comments. As part of the EMBO Press transparent editorial process, the point-by-point response is part of the Review Process File (RPF), which will be published alongside your paper. For more details on our Transparent Editorial Process, please visit our website: <https://www.embopress.org/page/journal/14693178/authorguide#transparentprocess>
You are able to opt out of this by letting the editorial office know (emboreports@embo.org). If you do opt out, the Review Process File link will point to the following statement: "No Review Process File is available with this article, as the authors have chosen not to make the review process public in this case."

4) a complete author checklist, which you can download from our author guidelines (<<http://embor.embopress.org/authorguide>>). Please insert information in the checklist that is also reflected in the manuscript. The completed author checklist will also be part of the RPF.

5) Please note that all corresponding authors are required to supply an ORCID ID for their name upon submission of a revised manuscript (<<https://orcid.org/>>). Please find instructions on how to link your ORCID ID to your account in our manuscript tracking system in our Author guidelines (<<http://embor.embopress.org/authorguide>>).

6) We replaced Supplementary Information with Expanded View (EV) Figures and Tables that are collapsible/expandable online. A maximum of 5 EV Figures can be typeset. EV Figures should be cited as 'Figure EV1, Figure EV2' etc... in the text and their respective legends should be included in the main text after the legends of regular figures.

- For the figures that you do NOT wish to display as Expanded View figures, they should be bundled together with their legends in a single PDF file called *Appendix*, which should start with a short Table of Content. Appendix figures should be referred to in the main text as: "Appendix Figure S1, Appendix Figure S2" etc. See detailed instructions regarding expanded view here: <<http://embor.embopress.org/authorguide#expandedview>>.

7) We would also encourage you to include the source data for figure panels that show essential data.

Numerical data should be provided as individual .xls or .csv files (including a tab describing the data). For blots or microscopy, uncropped images should be submitted (using a zip archive if multiple images need to be supplied for one panel). Additional information on source data and instruction on how to label the files are available <<http://embor.embopress.org/authorguide#sourcedata>>.

8) Regarding data quantification, please ensure to specify the name of the statistical test used to

generate error bars and P values, the number (n) of independent experiments underlying each data point (not replicate measures of one sample), and the test used to calculate p-values in each figure legend. Discussion of statistical methodology can be reported in the materials and methods section, but figure legends should contain a basic description of n, P and the test applied.

Please note that error bars and statistical comparisons may only be applied to data obtained from at least three independent biological replicates.

I look forward to seeing a revised version of your manuscript when it is ready. Please let me know if you have questions or comments regarding the revision.

Kind regards,

Deniz

Deniz Senyilmaz Tiebe, PhD
Editor
EMBO Reports

Referee #1:

The manuscript by Krueger et al. is a nice report on the function of Beta-Spectrin and the effect of potential actin-crosslinking on apical constriction in *Drosophila*. The authors show that formation of the ventral furrow is compromised after nanobody-driven disruption of beta-spectrin, and that beta-spectrin is enriched at apical, constricting surfaces. They further show that beta-spectrin appears to function downstream of twist and Rho signaling (apical Spectrin accumulation depends on twist function and ectopic activation of Rho can induce apical accumulation of Spectrin outside of the ventral furrow). The manuscript is fairly simple and straightforward, and the figures are very nicely done and very easy to read. There is a near absence of any tests of statistical significance, although the data looks convincing, but this is something for the authors to address on revision. In total, however, this manuscript would appear to be a good addition to EMBO Reports pending minor revisions, although this is a highly subjective threshold.

1) Much of the paper places Spectrin function in terms of cell ratcheting, but it isn't clear that this point is entirely proven. In the individual traces, there appears to be variable contractile behaviors. To show ratcheting defects occur, there should be quantifiable reversals in contraction. What are the number of expanding steps in WT and spectrin-defective embryos? What is the average contractile step size (are inwards contractions as large in spectrin-deficient embryos - from the traces it appears perhaps not?)? It appears that Figure EV1 partly addresses this, but I believe this is the summed pulsatile behavior, not the individual contractile and expansion components (the authors should feel free to clarify, however).

2) Is F-actin organization changed in spectrin-deficient embryos? This seems like an important point to show given spectrin's suspected function.

3) It would be good to take some of the key data points (contractile rates, cell area distributions, etc) and measure significance.

4) Many aspects of their data is also reminiscent of that observed in Miao et al., JCB, 2019 (large and small cells, defects in ratcheting), which may be worth commenting on and citing. The Introduction just says that ratcheting is poorly understood, but there has been work on this and should probably be cited - ratcheting and actin function has also been examined by Clement et al., Curr Biol, 2017 and Munjal et al., Nature, 2015, similarly Sumi et al., Dev Cell, 2019.

Minor notes:

a) Page 4, 2nd to last sentence, "It" is capitalized.

b) Figure 4D, there are a couple typos ("constrictin" and extra space)

Referee #2:

Krueger and colleagues investigate the mechanisms that stabilize pulses of apical constriction during the invagination of the *Drosophila* mesoderm. Inspired by the medioapical localization of bH-spectrin (an actin crosslinker) during pulses of contraction, the authors investigate the role that bH-spectrin may play in regulating medioapical networks. Nanobody-mediated bH-spectrin knock down did not affect cellularization, but caused mesoderm ingression defects, with irregular contraction reminiscent of the phenotype observed in *twi* mutant embryos. In *twi* mutants, mesoderm cells undergo pulses of contraction that are not stabilized. The authors showed that, in *twi* mutants, bH-spectrin does not accumulate medioapically, but remains junctional, consistent with its localization in the ectoderm. Optogenetic activation of Rho in the dorsal ectoderm caused the ectopic medioapical localization of bH-spectrin. Knock down of bH-spectrin did not affect the medioapical localization of myosin in the mesoderm, but led to networks that suffered tears as they contracted. bH-spectrin also affected the ability of some cells to stabilize pulses of apical constriction.

Pulsed contraction is present in a variety of morphogenetic processes, and therefore, the implication of bH-spectrin in pulse ratcheting is of broad interest to the cell and developmental biology communities. The paper is well conducted, using a variety of genetic, optogenetic, and protein-based disruption techniques. There are a couple of aspects that I find are somewhat underdeveloped, and I think they could relatively easily solidify the work:

1. I was a bit surprised the authors never looked at bH-spectrin together with actin reporters. What is the degree of colocalization between medioapical bH-spectrin and actin (it seems low for myosin based on Figure 1H-J)? How is the medioapical actin network in mesoderm cells affected by bH-spectrin knock down?

2. Figure 4C: the tears in the myosin networks that result from losing bH-spectrin are reminiscent of those observed in adherens junction mutants. Are adherens junctions disrupted when bH-spectrin is knocked down?

3. Figure 3M-Q: is the medioapical localization of bH-spectrin in response to Rho activation an

effect of biochemical or mechanical signaling? If the authors increased tissue tension using a different approach (e.g. by overexpression of a constitutively active Rok or Shroom, or by knocking down the myosin phosphatase) would that drive bH-spectrin to localize medioapically in the ectoderm?

4. The authors show that bH-spectrin localizes to the medioapical surface in mesoderm precursors, but is that accumulation also pulsatile, as shown for myosin before? Or is bH-spectrin loading at once onto the medioapical cortex at the onset of invagination (or linearly, or with some other dynamics)?

MINOR

1. Figure EV1 should be included as part of Figure 4.

TYPOS

1. Figure 1A: "adherence junction" should be "adherens junction"?

2. Page 6: "wild type embryos" should be "wild-type embryos".

3. Figure 4D: "constrictin" should be "constricting".

Rebuttal

We thank the reviewers for their suggestions and recommendations. Below, in blue our point-by-point response to the comments that have been raised. In the manuscript, new text is in red.

Referee #1:

The manuscript by Krueger et al. is a nice report on the function of Beta-Spectrin and the effect of potential actin-crosslinking on apical constriction in *Drosophila*. The authors show that formation of the ventral furrow is compromised after nanobody-driven disruption of beta-spectrin, and that beta-spectrin is enriched at apical, constricting surfaces. They further show that beta-spectrin appears to function downstream of twist and Rho signaling (apical Spectrin accumulation depends on twist function and ectopic activation of Rho can induce apical accumulation of Spectrin outside of the ventral furrow). The manuscript is fairly simple and straightforward, and the figures are very nicely done and very easy to read. There is a near absence of any tests of statistical significance, although the data looks convincing, but this is something for the authors to address on revision. In total, however, this manuscript would appear to be a good addition to EMBO Reports pending minor revisions, although this is a highly subjective threshold. First the authors adapted light controlled Rho1 activation at the membrane to increase locally contractility in a spatio-temporal controlled manner. This experiment is key to the study as it enables to conclude that there is temporal control of actin network's ability to contract.

1) Much of the paper places Spectrin function in terms of cell ratcheting, but it isn't clear that this point is entirely proven. In the individual traces, there appears to be variable contractile behaviors. To show ratcheting defects occur, there should be quantifiable reversals in contraction. What are the number of expanding steps in WT and spectrin-defective embryos? What is the average contractile step size (are inwards contractions as large in spectrin-deficient embryos - from the traces it appears perhaps not?)? It appears that Figure EV1 partly addresses this, but I believe this is the summed pulsatile behavior, not the individual contractile and expansion components (the authors should feel free to clarify, however).

As correctly pointed by this reviewr, the data presented in Figure EV5B (previous EV1) address the effect of β H-spectrin knockdown on the extent of cell ratcheting over a full pulsation cycle. In this revised version we performed an in-depth analysis of cell constriction and expansion (individually) which is presented in Figure EV5C,D (for all cell population) and Figure 4K (direct comparison between control cells and β H-spectrin knockdown cells). The new data is described in the results (p. 10): "Quantification of the constriction and expansion rates show that the constriction rate was reduced by ~24%, whereas the expansion rate increased by ~60% on average (Fig. 4K and Fig. EV5C,D), demonstrating that the change in the expansion rate is 2.5 times higher than the change in the constriction rate. Taken together these results are consistent with a role of β H-spectrin in apical ratcheting."

2) Is F-actin organization changed in spectrin-deficient embryos? This seems like an important point to show given spectrin's suspected function.

We generated a new fly line that allowed us to follow F-actin dynamics using a live reporter (LifeAct::mNeonGreen) in β H-spectrin knockdown embryos. We present the new data in Movie EV6 and describe this result on page 9: "However, over time the apical actomyosin network broke in cells that enlarged abnormally resulting in tissue tears (Fig. 4A-C) and local disruption of adherens junctions (Fig. EV3A-F). A similar phenotype was observed in β H-spectrin knock-down embryos expressing the F-actin reporter Life-Act::mNeonGreen (Movie EV6), suggesting that β H-spectrin is not required for initial assembly of the medio-apical actomyosin network but rather for maintaining its structural integrity as cells constrict."

3) It would be good to take some of the key data points (contractile rates, cell area distributions, etc) and measure significance.

We performed an ANOVA analysis for the relevant parameters and due to the big sample size we additionally analyzed the effect size by calculating the Cohen's d value to assess statistical significance. The statistical data were added to the respective Figure legends. Additionally, a new representation of the cell area distribution was added to the main figure (Figure 4J) and the pulsation rates of the different sample populations were analyzed in Figure EV 5C,D.

4) Many aspects of their data is also reminiscent of that observed in Miao et al., JCB, 2019 (large and small cells, defects in ratcheting), which may be worth commenting on and citing. The Introduction just says that ratcheting is poorly understood, but there has been work on this and should probably be cited - ratcheting and actin function has also been examined by Clement et al., Curr Biol, 2017 and Munjal et al., Nature, 2015, similarly Sumi et al., Dev Cell, 2019.

The references were added to the introduction and the following sentence was added to the discussion (p. 11): "Future work should also focus on how regulation of actin crosslinking interplays with other cellular processes involved in apical ratcheting, such as membrane turnover (Miao H et al JCB 2019)."

Minor notes:

a) Page 4, 2nd to last sentence, "It" is capitalized.

b) Figure 4D, there are a couple typos ("constrictin" and extra space)

Done.

Referee #2:

Krueger and colleagues investigate the mechanisms that stabilize pulses of apical constriction during the invagination of the *Drosophila* mesoderm. Inspired by the medioapical localization of β H-spectrin (an actin crosslinker) during pulses of contraction, the authors investigate the role that β H-spectrin may play in regulating medioapical networks. Nanobody-mediated β H-spectrin knock down did not affect cellularization, but caused mesoderm ingression defects, with irregular contraction reminiscent of the phenotype observed in *twi* mutant embryos. In *twi* mutants, mesoderm cells undergo pulses of contraction that are not stabilized. The authors showed that, in *twi* mutants, β H-spectrin does not accumulate medioapically, but remains junctional, consistent with its localization in the ectoderm. Optogenetic activation of Rho in the dorsal ectoderm caused the ectopic medioapical localization of β H-spectrin. Knock down of β H-spectrin did not affect the medioapical localization of myosin in the mesoderm, but led to networks that suffered tears as they contracted. β H-spectrin also affected the ability of some cells to stabilize pulses of apical constriction.

Pulsed contraction is present in a variety of morphogenetic processes, and therefore, the implication of β H-spectrin in pulse ratcheting is of broad interest to the cell and developmental biology communities. The paper is well conducted, using a variety of genetic, optogenetic, and protein-based disruption techniques. There are a couple of aspects that I find are somewhat underdeveloped, and I think they could relatively easily solidify the work:

1. I was a bit surprised the authors never looked at β H-spectrin together with actin reporters. What is the degree of colocalization between medioapical β H-spectrin and actin (it seems low for myosin based on Figure 1H-J)?

We co-stained embryos for β H-spectrin and phalloidin and presented the data in Figure 1K,L and Figure EV1 and described the new results on p. 6: "Phalloidin staining confirmed that β H-spectrin and F-actin colocalized in medio-apical fibers (Fig. 1K-L and Fig. EV1A-C)." Co-localization analysis revealed a Pearson's R value >0.7 (as mentioned in the figure legend of EV1).

How is the medioapical actin network in mesoderm cells affected by β H-spectrin knock down?

We generated a new fly line that allowed us to follow F-actin dynamics using a live reporter (LifeAct::mNeonGreen) in β H-spectrin knockdown embryos. We present the new data in Movie EV6 and describe this result on page 9: "A similar phenotype was observed in β H-spectrin knock-down embryos expressing the F-actin reporter Life-Act::mNeonGreen (Movie EV6), suggesting that β H-spectrin is not required for initial assembly of the medio-apical actomyosin network but rather for maintaining its structural integrity as cells constrict."

2. Figure 4C: the tears in the myosin networks that result from losing β H-spectrin are reminiscent of those observed in adherens junction mutants. Are adherens junctions disrupted when β H-spectrin is knocked down?

We analyzed adherens junctions by imaging the live reporter E-cad::mNeonGreen in β H-spectrin knockdown embryos. These new data are presented in the new figure EV3 and Movie EV5, and described in the results section (p. 9): “over time the apical actomyosin network broke in cells that enlarged abnormally resulting in tissue tears (Fig. 4A-C) and local disruption of adherens junctions (Fig. EV3A-F).”

3. Figure 3M-Q: is the medioapical localization of β H-spectrin in response to Rho activation an effect of biochemical or mechanical signaling? If the authors increased tissue tension using a different approach (e.g. by overexpression of a constitutively active Rok or Shroom, or by knocking down the myosin phosphatase) would that drive β H-spectrin to localize medioapically in the ectoderm?

We entirely agree with referee #2, that it is very interesting to elucidate whether β H-spectrin upregulation is due to biochemical or mechanical signaling. To address this question we blocked myosin-II activity by pharmacological inhibition of its pivotal activator the kinase ROCK using the inhibitor Y-27632 and performed optogenetic activation of Rho (Figure 3S,T and Figure EV2). We did not see upregulation of β H-spectrin in ROCK-inhibited embryos, suggesting that β H-spectrin upregulation depends on ROCK and myosin-II activity (i.e. mechanical signaling). However, this experiment does not completely rule out biochemical signaling as ROCK has targets others than myosin-II.

Therefore we performed a second experiment. We specifically induced mechanical stress at the apical cell surface and recorded the effect on β H-spectrin level in cells experiencing increased mechanical stress. To do so, we induced apical constriction in in three rows of cells using optogenetics leaving a gap of cells (1 or 2-3 cells wide) that remained non-activated but that experienced a stretching force due to the constricting (photo-activated) neighbors (see figure below). Cells that were immediately surrounded by constricting neighbors experienced maximal mechanical force, while cells that were mechanically buffered by non-activated cell neighbors experienced low mechanical force. This setup allowed us to observe a wide range of stretching force amplitudes at the same time. We observed that in none of the mechanically perturbed cells, β H-spectrin levels were different from non-activated cells outside the photo-activated region. While this result argues against a pure mechanical signaling pathway controlling β H-spectrin up-regulation, it is also possible that with our experimental set-up we do not reproduce the same type of mechanical stress generated normally by medio-apical actomyosin contractility. Given these concerns, this experiment does not allow us to conclusively distinguish between biochemical and mechanical signaling. Therefore we decided to not address this point in our manuscript, and only present the data on ROCK inhibition which allowed us to further dissect the signaling cascade controlling β H-spectrin. These new data are presented in Figure 3S,T and Figure EV2 and described in the results (p. 8): “Injection of the Rock inhibitor Y-27632 before optogenetic activation did not result in β H-spectrin or myosin-II upregulation. Thus, medio-apical β H-spectrin upregulation is dependent on the Rock-myosin-II branch of the Rho

signaling system (Fig. 3S,T and Fig. EV2A-I), consistent with the role of Rock in organizing the medio-apical actomyosin network.”

4. The authors show that β H-spectrin localizes to the medioapical surface in mesoderm precursors, but is that accumulation also pulsatile, as shown for myosin before? Or is β H-spectrin loading at once onto the medioapical cortex at the onset of invagination (or linearly, or with some other dynamics)?

To address this point, we performed a per-cell analysis of the β H-spectrin signal levels during ventral furrow formation during constriction and expansion cycles. These new data are presented in Figure EV 4 and described in the results section (p. 9): “Quantitative analysis of mVenus:: β H-spectrin signal over the course of apical constriction reveals that its levels progressively increased during the constriction phase and remained constant during relaxation (Fig. EV4A-F).”

MINOR

1. Figure EV1 should be included as part of Figure 4.

We decided to keep Figure EV5B (previous EV1) in the supplementary information, but we added new data in Figure 4K, which dissects the effect of β H-spectrin knockdown on cell pulsation with respect to constriction and expansion cycles in more details.

TYPOS

1. Figure 1A: "adherence junction" should be "adherens junction"?
2. Page 6: "wild type embryos" should be "wild-type embryos".
3. Figure 4D: "constrictin" should be "constricting".

Done.

Dear Stefano,

Thank you for submitting the revised version of your manuscript. It has now been seen by both of the original referees.

As you can see, the referees find that the study is significantly improved during revision and recommend publication. Before I can accept the manuscript, I need you to address some minor points below:

- Please discuss the point made by referee #2.
- Please provide 3-5 keywords for your study. These will be visible in the html version of the paper and on PubMed and will help increase the discoverability of your work.
- Our character limit for titles is 100 (including spaces) for technical reasons. Please shorten the title accordingly.
- As per our guidelines, please add a 'Data Availability Section', where you state that no data were deposited in a public database.
- We noted that EMBL is listed as a funder in the submission system, but not in the manuscript text.
- Movies need to be ZIPped with their legends. The legends should be removed from the Article file. The nomenclature should be 'Movie EV#'.
- The Appendix figure is should be named 'Appendix Figure S1'.
- Papers published in EMBO Reports include a 'Synopsis' to further enhance discoverability. Synopses are displayed on the html version of the paper and are freely accessible to all readers. The synopsis includes a short standfirst summarizing the study in 1 or 2 sentences that summarize the key findings of the paper and are provided by the authors and streamlined by the handling editor. I would therefore ask you to include your synopsis blurb.
- In addition, please provide an image for the synopsis. This image should provide a rapid overview of the question addressed in the study but still needs to be kept fairly modest since the image size cannot exceed 550x400 pixels.
- Our production/data editors have asked you to clarify several points in the figure legends (see attached document). Please incorporate these changes in the attached word document and return it with track changes activated.

Thank you again for giving us to consider your manuscript for EMBO Reports, I look forward to your minor revision.

Kind regards,

Deniz

--

Deniz Senyilmaz Tiebe, PhD
Editor
EMBO Reports

Referee #1:

I believe the authors have responded suitably to reviewer concerns, and the revised manuscript is

appropriate for publication in EMBO Reports.

Referee #2:

The authors have worked hard and addressed my concerns. I support publication of the manuscript. One completely optional suggestion for discussion is whether bH-spectrin is important to maintain the subcellular architecture of medioapical actomyosin networks (consistent with the medioapical localization of bH-spectrin), or the integrity of junctions that ensure supracellular connectivity (or both).

EMBL Heidelberg • Meyerhofstr. 1 • 69117 Heidelberg • Germany

Stefano De Renzis
MD, PhD
Group Leader
Developmental Biology Unit
T +49 6221 387-8109
F +49 6221 387-518

derenzis@embl.de

EMBL
Meyerhofstraße 1
69117 Heidelberg
Germany
www.embl.de

Heidelberg, 13-05-2020

Dear Deniz,

We have addressed all your points and referee #2 comment (please see page 10-11, “The function of β H-spectrin during apical constriction might be to cross-link and stabilize the medio-apical actomyosin network and/or to ensure its attachment to the junctions so that cells can maintain their constricted state also when neighboring cells constrict and exert forces on them.”)

We have entered the following keywords online:

Tissue morphogenesis; Actomyosin; Apical constriction; Tissue invagination; Optogenetics.

We have not found a place to upload the synopsis blurb, so we add it below:

The actin crosslinker β H-spectrin is up-regulated at the apical surface of invaginating mesodermal cells, where it localizes to the medioapical actomyosin network in a Twist/Rho-dependent manner. β H-spectrin functions as molecular ratchet to enable coordinated apical constrictions and tissue invagination.

Thank you for your help with processing our manuscript.

Best wishes,

Stefano De Renzis
On behalf of all the authors.

Dear Stefano,

Thank you for submitting your revised manuscript. I have now looked at everything and all looks fine. Therefore I am very pleased to accept your manuscript for publication in EMBO Reports.

Congratulations on a nice study!

Kind regards,

Deniz

--

Deniz Senyilmaz Tiebe, PhD
Editor

Corresponding Author Name: Stefano De Renzi and Daniel Krueger

Manuscript Number: EMBOR-2019-49858V1